# Recent Therapeutic Progress and Future Perspectives for the Treatment of Hearing Loss

**DOI:** 10.3390/biomedicines11123347

**Published:** 2023-12-18

**Authors:** Joey Lye, Derek S. Delaney, Fiona K. Leith, Varda S. Sardesai, Samuel McLenachan, Fred K. Chen, Marcus D. Atlas, Elaine Y. M. Wong

**Affiliations:** 1Hearing Therapeutics, Ear Science Institute Australia, Nedlands, WA 6009, Australia; joey.lye@earscience.org.au (J.L.); derek.delaney@earscience.org.au (D.S.D.); fiona.leith@earscience.org.au (F.K.L.); varda.sardesai@earscience.org.au (V.S.S.); marcus.atlas@earscience.org.au (M.D.A.); 2Centre for Ear Sciences, Medical School, The University of Western Australia, Nedlands, WA 6009, Australia; 3Faculty of Health Sciences, Curtin Health Innovation Research Institute, Curtin University, Bentley, WA 6102, Australia; 4Ocular Tissue Engineering Laboratory, Lions Eye Institute, Nedlands, WA 6009, Australia; samuelmclenachan@lei.org.au (S.M.); fred.chen@lei.org.au (F.K.C.); 5Centre for Ophthalmology and Visual Sciences, The University of Western Australia, Nedlands, WA 6009, Australia; 6Vitroretinal Surgery, Royal Perth Hospital, Perth, WA 6000, Australia; 7Ophthalmology, Department of Surgery, University of Melbourne, East Melbourne, VIC 3002, Australia; 8Centre for Eye Research Australia, Royal Victorian Eye and Ear Hospital, East Melbourne, VIC 3002, Australia; 9Curtin Medical School, Faculty of Health Sciences, Curtin University, Bentley, WA 6102, Australia

**Keywords:** hearing loss, therapeutics, inner ear, hair cell, gene therapy, cell therapy

## Abstract

Up to 1.5 billion people worldwide suffer from various forms of hearing loss, with an additional 1.1 billion people at risk from various insults such as increased consumption of recreational noise-emitting devices and ageing. The most common type of hearing impairment is sensorineural hearing loss caused by the degeneration or malfunction of cochlear hair cells or spiral ganglion nerves in the inner ear. There is currently no cure for hearing loss. However, emerging frontier technologies such as gene, drug or cell-based therapies offer hope for an effective cure. In this review, we discuss the current therapeutic progress for the treatment of hearing loss. We describe and evaluate the major therapeutic approaches being applied to hearing loss and summarize the key trials and studies.

## 1. Introduction

### 1.1. Clinical Burden of Sensorineural Hearing Loss

The World Health Organization (WHO) has estimated over 1.5 billion people worldwide currently experience some form of hearing loss (HL), which is expected to increase significantly by 2050 [1]. Hearing impairment is a disabling disorder which can significantly affect livelihood and is a leading global public health problem. The economic burden of HL has been estimated to cost $980 billion USD annually to provide healthcare, societal and educational support to patients [1]. A study reported by Mohr et al. [2] suggested that an individual living in the United States is expected to spend $297,000 USD over their lifetime on hearing health. However, a recent study suggested the cost of hearing-related health expenditures may exceed $1 million USD for at-need children to have access to special education, counselling and hearing interventions [3].

People with HL can experience significant psychosocial impacts throughout their lives. When HL is unaddressed in early life, educational outcomes and the development of social skills are impacted, which can lead to reduced employment opportunities and quality of life in adulthood [4,5]. Most cases of HL are preventable but limited awareness and delayed self-diagnosis contribute to the neglect of early intervention and preventative measures. Furthermore, the prevalence of HL is known to increase with age, with age-related hearing loss (ARHL) or presbycusis being a common problem for older people. In recent years, there have been increased reports of a significant association of cognitive impairment and ARHL [6,7]. Dementia has been shown to be up to five times more prevalent among older people with moderate to severe HL than those with normal hearing [8].

Sensorineural hearing loss (SNHL) is the most frequent form of HL and is defined by the inability to transmit incoming soundwaves and perceive them as sound due to pathophysiological changes in the auditory pathway. This is defined by the degeneration of cochlear hair cells (HCs) and/or afferent neurons and sounds are perceived as muted and distorted compared to other forms of HL. SNHL is progressive and results in permanent and irreparable damage. SNHL can be inherited or acquired through age, genetics and environmental factors, which can also interact to cause unilateral or bilateral HL [9].

### 1.2. Cause of Sensorineural Hearing Loss

The cochlea is responsible for the transduction of sound to the brain and the sensory cells of the organ of Corti within it facilitate this process. The organ of Corti houses inner hair cells (IHCs) and outer hair cells (OHCs), which are responsible for the primary transduction of sound through spiral ganglion nerves (SGNs) and mediating oscillatory forces, respectively [10] (Figure 1). HCs are named for the “hair bundle” on their apical surface, which is comprised of actin-based projections called stereocilia. Stereocilia are embedded in the tectorial membrane, through which they detect soundwaves originating from the middle ear [11]. Soundwaves cause the tectorial membrane to vibrate, which moves stereocilia and triggers the opening of mechanoelectrical transduction channels, causing depolarization of the SGNs and subsequently, transmission of the sound signal to the brain [11,12,13,14,15]. Moreover, HCs are surrounded by a heterogenous population of various types of supporting cells (SCs), which are thought to play a role in potassium ion regulation, osmotic balance and limited epithelial repair following trauma (Figure 1).

SNHL is a result of damage to or malfunction of the organ of Corti that interferes with the process of sound transduction. SNHL can be acquired or hereditary. Acquired SNHL is largely environmental and is a result of cumulative damage to the organ of Corti over the course of life [16]. Noise exposure from industrial work or loud music are common causes of noise-induced hearing loss (NIHL). Drug-induced HL is a significant clinical problem as ototoxicity is a prevalent side effect, with a review by Rizk and colleagues in 2020 listing 194 medications of various classes having ototoxic side effects [17]. Some of these medications that are prescribed for long-term management of chronic disorders, such as gentamicin or cisplatin, can cause severe and irreversible damage to sensory cells in the cochlea.

ARHL, or presbycusis, and NIHL will become increasingly prevalent with the extending human lifespan and increased use of personal listening devices [1]. Bilateral SNHL currently affects over 65% of adults over 65 years of age [16]. Additionally, smoking and cardiovascular and metabolic disorders have been shown to be associated with SNHL. The causal mechanisms are still to be established for these associations, however it is thought for cardiovascular and metabolic disorders that damage to the cochlear vasculature may play a role [18,19,20,21].

Hereditary SNHL affects around 1 in every 1000 newborns and can be devastating with genetic mutations to single genes, resulting in complete or progressive deafness in some diseases [22,23]. Many of the genes affected in disorders of hereditary SNHL encode the essential components of HCs and mechanoelectrical transduction. For example, *GJB2* encodes the gap junction protein connexin 26, while many of the genes in Usher syndrome such as *MYO7A* and *WHRN* encode for essential hair bundle components [24,25]. Mutations in these genes severely affect signal transduction, as unbalanced ion concentrations and destabilized stereocilia render HCs insensitive, or unresponsive in worst cases, to sound and result in deafness. There are currently 124 genes that have been implicated in various forms of non-syndromic hereditary SNHL (https://hereditaryhearingloss.org/).

### 1.3. Current Interventions against Sensorineural Hearing Loss

The most common intervention for patients with SNHL is using prosthetics such as hearing aids or cochlear implants to aid hearing. While these are effective for less severe forms of SNHL, they do still require some HC functionality for sound transduction [26]. Moreover, they are currently unable to mimic the quality of natural hearing.

There is a severe paucity of small molecule drugs available for treating SNHL and these are known to have varying degrees of effectiveness. The corticosteroid dexamethasone is commonly administered either orally or intratympanically for its anti-inflammatory properties and has shown some benefit [27,28]. Unfortunately, dexamethasone has poor bioavailability in the cochlea due to limited uptake in the inner ear when administered orally and is readily cleared from the cochlear fluids when injected intratympanically [29]. Recently, SPT-2101, a hydrogel formulation containing dexamethasone, has been developed by Spiral Therapeutics (South Francisco, CA, USA) and is currently undergoing a Phase I clinical trial in Australia (ACTRN12621000964819), aiming to provide sustained release in the inner ear. For drug-induced HL, sodium thiosulfate was recently approved by the US Food and Drug Administration (FDA) to prevent cisplatin-induced HL in patients undergoing cisplatin chemotherapy [30]. Sodium thiosulfate directly chelates cisplatin, however, which risks interfering with the chemotherapeutic effects [31]. A clinical trial of intratympanically administered sodium thiosulfate, using a hydrogel-based formulation for controlled release (DB-020, Decibel Therapeutics, Boston, MA, USA), recently completed Phase I clinical trials (Table 1). Intravenously administered sodium thiosulfate remains the only novel drug to be approved by the US FDA in 20 years of research in hearing therapeutics [32].

## 2. Novel Therapeutic Approaches to Restore Hearing

While the current options of drug treatment for any form of SNHL are limited, there are many novel treatments undergoing clinical trials. As corticosteroid therapy is proving to be insufficient for widely treating SNHL, novel drugs targeting other pathways are being investigated. One pathway which has recently seen great research interest and investment is regenerating HCs through transdifferentiation of SCs. Small molecule and genetic approaches are being developed to induce the signaling pathways to initiate HC transdifferentiation. Moreover, genetic approaches are being increasingly investigated to treat the most severe hereditary forms of SNHL where, for example, gene supplementation or editing of an endogenously faulty gene could restore HC function. In the following sections, we will review the recent therapeutic strategies for targeting the pathways that promote the restoration of the sensory epithelium and SGNs.

## 3. Emerging Drug Therapies

As with other highly specialized tissues, the development of the mammalian inner ear is guided by the precise spatiotemporal activation and inhibition of key signaling pathways. These key signaling pathways include Wnt, which interacts with Sonic Hedgehog signaling to specify the apical–basal polarity of the cochlea and cochlear outgrowth, and Notch, which is involved in cell fate specification and patterning within the organ of Corti [33]. Several groups have demonstrated that in vitro manipulation of these signaling pathways can induce differentiation of embryonic and induced pluripotent stem cells into HC-like cells [34,35,36,37].

Most recently, Zheng-Yi Chen’s group was able to demonstrate a novel approach to reprogram SCs into HCs in adult mice using small molecule drugs. Quan et al. [37] demonstrated the transdifferentiation of primarily interdental cells into OHCs in adult mouse cochlear explants using a cocktail treatment consisting of valproic acid, siRNAs activating c-Myc, lithium chloride as a Wnt activator and a cyclic AMP agonist Forsklin. Single-cell RNAseq of the transformed HC-like cells showed that the cocktail stimulated a direct transition from an interdental cell type to that resembling fetal HCs, showed by the co-staining of the HC markers with SOX2 [37]. Moreover, this study achieved an additional major milestone by demonstrating transdifferentiation in the adult mouse cochlea, whereas most studies to date have been executed on neonatal mice.

A few small molecule drugs attempting to induce HC differentiation have recently undergone clinical trials for treating various forms of SNHL in humans. One is FX-322 from Frequency Therapeutics, which is an intratympanically injected formulation of the Wnt activator, CHIR99021, and histone deacetylase inhibitor, valproic acid. FX-322 just completed Phase IIb clinical trials, showing no statistically significant difference at day 90 compared to the placebo. The other drugs are LY3056480 and PIPE-505, from Audion Therapeutics (Amsterdam, The Netherlands) and Pipeline Therapeutics (San Diego, CA, USA), respectively, which are both γ-secretase inhibitors. γ-secretase plays a key role in Notch signaling by cleaving the intracellular domain of Notch and initiating the signaling pathway. Moreover, in the adult mammalian cochlea, Notch-mediated lateral inhibition prevents transdifferentiation of HCs [38,39]. At this stage, both studies appear to be either on hold or withdrawn.

Cellular quiescence occurs via the inhibition of cyclin/cyclin-dependent kinase (CDK) and the hypophosphorylation of retinoblastoma protein [40]. CDK inhibitors, such as p27^Kip1^, are expressed in the cochlea at a level sufficient enough to promote quiescence of the HCs [41,42,43]. The deletion of p27^Kip1^ in transgenic mice resulted in the proliferation of SCs and the regeneration of HCs [44,45,46]. The HC-specific conditional deletion of p27^Kip1^ in neonatal mice resulted in the proliferation and improved survival of HCs without any adverse effects on hearing, as measured by the auditory brainstem response (ABR). IHCs in particular were more proliferative than OHCs in response to the knockout of p27^Kip1^ [43]. These findings suggested that p27^Kip1^ could represent a good therapeutic target to minimize cell death and promote HC regeneration in the cochlea. Indeed, Sound Pharmaceuticals (Seattle, WA, USA) is conducting a preclinical study of SPI-5557, a drug that inhibits p27^Kip1^, with the aim of regenerating cochlear HCs [47].

Multiple strategies have been suggested to deliver pro-neurogenic growth factors to the inner ear to promote SGN growth and regeneration. Neurotrophins, such as the brain-derived neurotrophic factor or neurotrophin-3, have been demonstrated to have protective effects in animals exposed to ototoxic insults [48]. Moreover, they have also been shown to promote neural outgrowth and the formation of synapses, making them and their Trk receptors promising candidates for the treatment of SNHL [49,50]. Small molecule Trk receptor agonists have been proposed as a strategy to induce neuroprotective pathways and improve the pharmacokinetics of Trk receptor activation in the cochlea [49,51,52]. A recent study by Fernandez et al. [53] demonstrated the protective and regenerative effects of the TrkB agonists amitriptyline and 7,8-dihydroxyflavone on an animal model of glutamate excitotoxicity, which damages SGN terminal processes. Another recent study demonstrated that a neurotrophin-3 mimetic, 1Aa conjugated with risedronate, promoted the regeneration of cochlear ribbon synapses and neurite outgrowth in mouse cochlear explants [54]. An alternative approach for introducing neurotrophins into the cochlea has additionally been proposed through gene delivery to the cochlea [55]. We will more broadly discuss gene therapy in the following section.

While small molecule drugs are more traditional, stable and less complex than the newer therapies discussed in the subsequent sections, a major issue for therapy lies in drug delivery. The inner ear is among the hardest areas in the body to deliver drugs to, as systemically administered drugs are poorly taken up and the cochlea itself lies behind the tympanic and RWM when administering via the ear canal. Recent developments in hydrogel and nanotechnology have proposed to provide a reservoir of continually delivered drug and increase permeation. This would theoretically solve the problem of controlled release and continual dosage [56,57,58,59].

Although this review largely focuses on HC regeneration, there are many other small molecule inhibitors undergoing clinical trials. These are aimed at a range of SNHL etiologies, including sudden SNHL and cytomegalovirus-induced SNHL. While this review is more focused on applications for the more prevalent forms of acquired and hereditary SNHL, we have compiled a table of all the small molecule drugs currently undergoing clinical trials that is registered in the National Library of Medicine’s (NLM) database (www.clinicaltrials.gov) for other forms of SNHL, as shown in Table 1.

## 4. Gene Therapy

Gene therapy generally refers to the supplementation of an endogenously healthy gene and offers a prospective solution for certain forms of SNHL. In a landmark study, ATOH1 (also known as MATH1 for “mouse Atoh1” and HATH1 for “human ATOH1”), a basic helix-loop-helix transcription factor, was identified to be essential in the generation of inner ear HCs and was able to enhance HC regeneration when overexpressed [60,61]. *ATOH1* has since been used as a prime candidate in HC regeneration research to varying degrees of success in published studies. Guinea pigs treated with *ATOH1* gene therapy showed a significant increase in the number of HCs marked by a positive myosin VIIa (MYO7A) expression but failed to regenerate a full complement of ribbon synapses [62]. Overall, the hearing in these guinea pigs did not recover, suggesting that *ATOH1* gene therapy alone is unable to convert non-sensory cells into HCs, and is not capable of rescuing the phenotype of surviving HCs [62]. Apart from *ATOH1*, other studies have looked at the candidate signaling pathways that control the *ATOH1* gene expression, such as Notch signaling, which plays a key role in HC and SC differentiation, and Wnt/β-catenin signaling pathways.

A popular option for delivering gene therapy is using viral vectors, such as lentiviruses and adeno-associated viruses (AAVs). While lentiviral vectors have a greater packaging capacity than AAV vectors, ~7.5 kb versus ~5 kb, respectively, the genes that encode HC mechanotransduction components, commonly mutated or lost in genetic disorders such as Usher syndrome (USH), often exceed even the lentiviral packaging capacity.

The AAV delivery of full-length *MYO7A* cDNA has been shown to be effective in vivo and in vitro and resulted in wild-type levels of expression [63,64,65,66]. Packaging full-length *MYO7A* cDNA into AAVs produced vectors with heterogeneous, fragmented genomes (fAAVs) that were able to reconstitute full-length *MYO7A* cDNA post-infection. However, fAAVs are relatively ineffective and unable to be definitively characterized, so are unacceptable for clinical translation [65]. As *MYO7A* is relatively large, it exceeds the 5 kb capacity of a single AAV, so a dual AAV vector platform might overcome this limitation [64,65,66]. An alternative strategy to overcome the size limitations reported for *OTOF* (otoferlin) and *PCDH15* (protocadherin 15) has been to decrease the gene size by deleting redundant domains, which has shown some success in mouse models [67,68].

Dual AAV vectors are designed as pairs, containing either the 3′ or 5′ fragments of *MYO7A* cDNA with an overlapping region. These AAVs can then be reconstituted by either splicing (trans-splicing), homologous recombination (overlapping) or a combination of the two methods (hybrid) [66]. Dual AAV treatments have transduced retinal pigment epithelium and photoreceptor cells, resulting in the expression of a functional full-length myosin VIIA protein and the rescue of the mutant phenotypes, which is suggestive of the successful homologous recombination of *MYO7A* gene fragments. However, this method had low efficiency and minimal overexpression of *MYO7A* in infected cells [64].

A breakthrough in rescuing HL and vestibular function has been achieved in adult mouse models via a synthetic designer AAV vector (AAV2/Anc80L65) [69]. The injection of AAVs into the posterior semicircular canal of adult mice has demonstrated efficient transduction of all IHCs, a proportion of OHCs, as well as all the SCs in the vestibular system. These findings indicated the potential of this viral vector and canalostomy as a surgical approach for gene therapy in cochlear and vestibular sensory organs [69].

In another study, AAV2/Anc80L65 injected in neonatal *Ush1c* c.216G>A mice revealed an improvement in IHC and OHC survival, with the HCs remaining mechanosensitive during the first postnatal week [70]. Using a similar approach, the injection of AAV2/1 vectors in *Tmc1* mutant mice, known as Beethoven (*Bth*), targeted 80–90% of IHCs and yielded moderate auditory rescue [71].

The direct injection of AAV serotype 2/8 to deliver *DFNB31* (also known as *WHRN*) cDNA through the inner ear successfully infected numerous HCs, where the localization of whirlin was observed. In the cells where whirlin was expressed, an elongation of the stereocilia bundles was initiated, resulting in the restoration of stereocilia to a wild-type length, which is critical for IHCs to be functional. This whirlin gene therapy approach was able to successfully restore balance and auditory function in Whirler mice for approximately four months [72].

Recombinant AAV vectors, capsid serotype 2 or 8, containing *Clrn1* cDNA and driven by a small chicken β-actin (smCBA) promoter, were constructed and injected directly into the inner ear through the RWM of KO-TgAC1 mice to restore the *Clrn1* expression [73]. Following a single in vivo injection of the AAV2/8-*Clrn1*-UTR vector into the cochlea on P1–P3, the KO-TgAC1 mice displayed normal hearing through to adult life and the preservation of the hair bundle structure. Transfection with AAV-*Clrn1*-UTR successfully suppressed the progressive HL phenotype displayed in KO-TgAC1 mice, offering a potential new gene therapy for treating USH3A [73].

*Ush1g* knockout mice, which lacked *Sans*, displayed no ABR to sound, indicating complete deafness, as well as severe vestibular dysfunction [74]. A single dose of AAV8 containing a replacement *Ush1g* gene delivered locally into the inner ear of newborn knockout mice completely restored the vestibular function to wild-type levels and restored hearing of low frequency loud sounds (≥75 dB) [74]. Gene replacement therapy is quite promising; however, the applicability of this for HL treatment is still challenging due to the characteristics and anatomy of the inner ear structure [75].

Akouos, Inc. has several ongoing development programs involving AAV development and gene therapy. The most advanced program is AK-OTOF, which aims to deliver the *OTOF* gene to cochlear HCs in patients with hereditary SNHL resulting from mutations in the aforementioned gene [48]. AK-OTOF is currently undergoing Phase I–II clinical trials. Other gene therapies developed by Akouos, Inc. (Boston, MA, USA), including *CLRN1* and *GJB2*, are in preclinical development (Table 2).

A current limitation of AAVs is their transfection efficiency in vivo. While AAVs are able to cross the RWM, their ability to reach the cochlea may be hampered by anti-AAV antibodies and clearance via the cochlear aqueduct [76]. György et al. [77] designed an approach to increase transfection efficiency using exosomes to encapsulate AAV1, which resulted in increased transfection efficiency in vivo compared to AAV1 alone and a partial rescue of hearing in *Lhfpl5*^−/−^ mice. Moreover, off-target effects and toxicity in the central nervous system are possible by passive transport from the cochlear aqueduct [55,78].

Non-viral vectors are an alternative gene delivery method to viral vectors and largely refer to nanoparticles. Nanoparticles are small and advantageous as drug carriers or gene vectors due to their ability to be modified and functionalized with various excipients, such as lipids or polymers like PEG (polyethylene glycol), PLGA (poly(lactic-co-glycolic acid)) and alginate among others, to improve permeability and targeting. Their main advantage over AAVs is that they can be formulated to have larger carrying capacities, allowing them to carry and deliver larger genes and even proteins. Gene transfection with nanoparticles tends to be less durable than with AAV, although this may not be a disadvantage where the prolonged expression of an exogenous gene may cause unwanted effects [79]. We and others have recently reviewed how nanoparticles are formulated for various therapeutic applications to the inner ear, which we would recommend to the interested reader as that is beyond the scope of this review [57,80]. Briefly, nanoparticles can largely be categorized as lipidic or inorganic, with the former being the more established for general nanomedicine. There are general limitations with nanoparticles in their variable biodegradability, consistency of manufacture and less durable transfection than AAVs. However, an ideal formulation and delivery mode is yet to be developed and recent innovations in nanotechnology and drug delivery put an ideal nanoparticle for inner ear drug delivery within reach. In the following sections on CRISPR and RNAi, we discuss some preclinical studies that have employed lipid nanoparticles for drug delivery.

## 5. Gene Editing Therapies

### 5.1. CRISPR-Cas Therapeutics

The CRISPR-Cas9 method of genome editing has great potential to be used for treating certain forms of SNHL. CRISPR, or Clustered Regularly Interspaced Palindromic Repeats, is currently the most prominent method for gene editing. The CRISPR system has two components: a Cas nuclease and guide RNA (gRNA), which guides the nuclease to the target area in the genome [81]. The target area is required to be adjacent to a protospacer-adjacent motif (PAM), a short 3–6 bp sequence found throughout the genome. However, recent variants have been identified that recognize much wider PAM sequence varieties and have been dubbed “near PAM-less” [82,83]. Once guided to the target area, the Cas nuclease causes a double-stranded break in the DNA, where modifications can be made alongside the repair of the break [84,85].

CRISPR-mediated genome editing is especially applicable for certain genetic hearing disorders. Many genetic hearing disorders arise from single mutations in the DNA sequence of a gene, which can theoretically be rectified using CRISPR. As the target cell types in the cochlea are terminally differentiated, many of the strategies described later involve disrupting the target allele of an autosomal dominant disorder or using base editing to modify and repair single nucleotides. Like other methods of repair, notably homology-directed repair, this requires cell division to occur to introduce the modification [86]. At this stage, CRISPR-mediated therapy for some forms of genetic HL is in early development; however, more work has recently been produced in this area, which we have summarized in Table 3.

Gao et al. [87] used a CRISPR approach to disrupt the *Tmc1* mutation in the *Bth* mouse model of autosomal dominant HL. The study employed *Streptococcus pyogenes* Cas9 (SpCas9, one of the most widely used experimentally) to target and disrupt the mutant *Tmc1^Bth^* allele in neonatal mice, using liposomes to deliver the gRNA-CRISPR-Cas9 system into the mouse scala media. The CRISPR-mediated therapy mitigated the progressive cochlear HC death exhibited by the *Bth* model compared to un-injected mice. The treated *Bth* mice also exhibited improved low–middle frequency ABR thresholds and an increased startle response compared to the un-injected mice [87]. A more recent study by György et al. [88] used a PAM variant of the *Staphylococcus aureus* Cas9 (SaCas9-KKH), which was able to target the *Bth* mutation in the mice and the human p.M418K mutation in human cell lines. The effects of this treatment were sustained for longer than the Gao et al. [88] study. Finally, Zheng et al. [89] used an RNAi approach with CRISPR-CasRx, which is an attractive approach as it has a decreased risk of off-target effects, with transcripts being targeted rather than the genome, resulting in improved hearing up to 15 weeks post-treatment. Similar studies have been conducted using SpCas9 and SaCas9-KKH on mouse models of *KCNQ4*-associated autosomal dominant hearing loss [90,91].

A strategy for gene editing by homology-mediated end joining (HMEJ) was recently presented by Gu et al. [92]. A mutation in the *Klhl18* (Kelch-like family member 18) gene led to the dysfunction of IHCs, which resulted in abnormal ABR but normal DPOAE thresholds [93]. AAV9 and AAV-PHP.eB were used to package the CRISPR-Cas9 knock-in system for delivery into the *Klhl18^lowf^* mice, which modeled the autosomal recessive condition in humans. Using the HMEJ strategy, they were able to rescue IHC stereocilia morphology eight weeks after the injection of the dual AAV system and demonstrated improved auditory function up to six months post-treatment [92].

### 5.2. Base Editing Therapeutics

While the sequence-targeting ability of the Cas nuclease is highly useful, double-stranded breaks are not usually practical for the correction of mutations, especially for single base point mutations where the swapping of one nucleotide pair is sufficient. Double-stranded breaks induced by Cas additionally induce DNA repair mechanisms that may either create insertion/deletion polymorphisms (indels), leading to off-target effects, or the reintroduction of the original pathologic variant [94,95]. The most notable innovation in CRISPR technology for overcoming this is the introduction of base editors, which have recently been tested in preclinical models of genetic SNHL. Base editors are able to modify individual base pairs in double-stranded DNA without causing a double-stranded break. They are chimeric proteins consisting of a modified Cas nuclease, where one of the catalytic domains is inactivated, and a single-stranded DNA deaminase enzyme [84,96]. Base editors are advantageous for gene editing HCs and SCs as mitosis is not required to complete changes to the genome [97]. There are two main classes of base editors: cytosine base editors, which convert C•G base pairs to A•T, and adenosine base editors, which catalyze the inverse A•T to C•G conversion.

Two papers by Yeh and colleagues were the first to demonstrate the potential of base editing in inner ear therapy. The first sought to modify the β-catenin protein to prevent its degradation by the proteasome. Β-catenin is a key effector protein in the Wnt signaling pathway, a crucial pathway regulating HC development, and is targeted for degradation upon the phosphorylation by GSK3β. The study by Yeh et al. [98] showed that the modification of β-catenin prevented phosphorylation and degradation and prolonged Wnt signaling in vitro. Moreover, the authors used a lipid-mediated approach for delivery, as in Gao et al. [87], and showed specific gene editing in SCs when the base editor was applied in vivo. The second paper showed some improvement in hearing when a base editing approach was applied to treat *Baringo* mice, which had an autosomal recessive form of HL caused by a T•A-to-C•G mutation leading to a non-functional TMC1 protein [99]. This particular approach additionally highlighted the advantage of base editing in treating autosomal recessive disorders by being able to specifically correct the mutation in both alleles in terminally differentiated cells.

Very recently, base editing was applied to an animal model of USH type 1 [100]. This study marked the first in vivo application of CRISPR for USH. The authors generated a mouse model of USH1F, first using CRISPR to introduce the R245X (c.733C•T) mutation, which produced a stop codon in place of an arginine, resulting in a truncated Pcdh15 protein and disrupted hair bundles. The authors used a dual AAV vector with a split intein system to deliver adenine base editors to the cochlea of *Pcdh15^R245X/R245X^* mice at P0–P1 and a late-stage conditional knockout model where *Pcdh15* was deleted at P15.5 under the control of the *Myo15*-Cre promotor. The adenine base editors were successful in correcting the C•T mutation for A•G in vivo. However, they were unable to restore hearing in the *Pcdh15^R245X/R245X^* mice [100]. The base editor was able to restore hearing in the late-stage conditional knockout model, indicating that the therapy may potentially be more successful if administered in the prenatal period.

### 5.3. CRISPR in Human Models of SNHL

While animal models have been useful for disease modeling and the functional evaluation of therapeutics, human phenotypes are not always replicated, usually due to polymorphisms of protein-coding genes. This is particularly relevant for assessing the effect of CRISPR on the editing of causative genes where certain variants may lead to different phenotypes in animals and humans. As human tissue samples are scarce, in vitro cell-based models are used for convenience and regenerative capacity and are sufficient for investigating gene editing at the molecular level. Patient-derived cell models have the additional benefits of being relatively easy and non-invasive to establish and numerous studies have reprogrammed various patient cell types into induced pluripotent stem cells (iPSCs) [101,102,103,104,105].

There are numerous studies that have generated cell models from patients with various subtypes of USH-associated genes and used CRISPR to correct the mutation, which we have summarized in Table 3. A notable study by Tang et al. [101] investigated the use of CRISPR-Cas9 to correct the *MYO7A* mutation in cells derived from a patient with profound HL. The authors generated three iPSC lines. The first cell line was a compound heterozygous mutant from the patient with two *MYO7A* mutations, c.1184G>A and c.4118C>T, and the second was generated from the asymptomatic father of the patient with only the *MYO7A* c.1184G>A mutation. The final cell line was derived from a normal donor. The authors used CRISPR-Cas9 and a HDR template to correct the c.4118C>T mutation in the iPSCs and then differentiate them into HC-like cells. A comparison of the corrected patient cell line showed a similar stereocilia bundle morphology to the control and asymptomatic cells, while the unedited patient cell line had curved, disconnected and shorter stereocilia. The corrected patient cell line additionally displayed a comparable electrophysiological response to the asymptomatic and patient cell lines [101].

Similarly, Sanjurjo-Soriano et al. [105] reported the successful seamless correction of the two most prevalent *USH2A* mutations (c.2276G>T or c.2299delG) in USH2A patient-derived iPSCs via a CRISPR-Cas9 system. The iPSCs were corrected using the high target efficacy and specificity of eSpCas9, without any off-target mutagenesis identified in the corrected iPSCs. These corrected iPSCs retained their pluripotency characteristics and genetic stability. Furthermore, they identified aberrant mRNA levels associated with both mutations that reverted to wild-type following the CRISPR-Cas9 correction, uncovering a novel mechanism for c.2299delG mutation pathogenesis. Another study demonstrated the successful editing of USH2A patient-derived fibroblasts containing the 2299delG mutation [102].

Human cell models have not been limited to USH. Recently, a study applied CRISPR to patient-derived cells with a mitochondrial 12S rRNA 1555A>G and *TRMU* c.28G>T allele mutation, which has been associated with non-syndromic HL and an increased susceptibility to aminoglycoside-induced HL [106]. This patient had non-syndromic HL and the authors sought to investigate whether a disruption of the *TRMU* c.28G>T allele, which interacts with the 12S rRNA 1555A>G mutation, would improve the functional properties in the cell model. The authors generated cell lines from the patient, an asymptomatic family member with only the 12S rRNA 1555A>G mutation and an unrelated control, corrected the *TRMU* c.28G>T allele using CRISPR-Cas9 and then differentiated the cells into HC-like cells to assess their function. The corrected cells displayed more robust differentiation into HC-like cells with a comparative stereocilia morphology, electrophysiological function, mitochondria count and HC-specific gene expression to the cells of the asymptomatic family member [107].

Another study generated iPSCs from the dermal fibroblasts of a patient with profound HL who had compound heterozygous mutations c.4642G>A and c.8374G>A in the *MYO15A* gene, which is the third most prevalent congenital HL gene [108]. iPSCs were also generated from their normal hearing father, who had the c.4642G>A mutation, and a healthy donor. These iPSC lines were differentiated into HC-like cells. The patient-derived cell line exhibited severely retarded stereocilia formation after differentiation and eventually formed syncytia or died. CRISPR-Cas9 with a HDR template was used to correct the c.4642G>A mutation in the patient-derived iPSCs, which resulted in restored stereocilia formation and no syncytia being formed [109].

CRISPR-Cas technology is continually being improved with innovations in protein engineering and evolution. The newly established prime-editing technique, for example, improves upon base editing by having the ability to install a greater variety of single base changes in the genome, which is highly applicable to autosomal recessive forms of genetic SNHL. While the existing concerns of off-target genome editing remain, the more clinically relevant concerns are when to introduce CRISPR-Cas therapy. For example, if genome editing intervention at the fetal stage is practicable given the terminal development of the neonatal inner ear, would the ideal delivery vehicle be a viral or nanoparticle vector? Moreover, given the promising applications of CRISPR for treating genetic forms of HL and the existing use of patient-derived cell lines, patient screening and personalized medicine approaches are critical for timely intervention with future CRISPR-based therapies.

**Table 3 biomedicines-11-03347-t003:** Therapeutic applications of CRISPR-Cas and related systems for SNHL.

Gene/Disease Model	Model	Type of Edit	Delivery Mode	Effect	Reference
*DFNA36*	*Bth* mice, P1–P2	Allele disruption by SpCas9	Lipid nanoparticle	Decreased OHC death compared to untreated; improved low–middle frequency hearing	[87]
*DFNA36*	*Bth* mice, P1–P2	Allele disruption by SaCas9-KKH	Anc80L65	Preservation of low–middle frequency hearing and decreased HC death; measured up to 24 weeks	[88]
*DFNA36*	*Bth* mice, P1–P2	mRNA transcript disruption by CRISPR-CasRx	Anc80L65	Improved hearing, HC survival and hair bundle formation; measured up to 15 weeks	[89]
*KCNQ4*	Knock-in mouse model	SpCas9	Lipid nanoparticle or AAV2/Anc80L65	Improved auditory function; OHC death not significantly abrogated; no difference in efficacy with injection routes	[91]
*KCNQ4*	*KCNQ4^G229D^* mice, P1–P2	Allele disruption by SaCas9-KKH	AAV-PHP.eB	Improved auditory function compared to untreated, increased survival and hyperpolarized resting potentials of OHCs; measured up to 12 weeks	[90]
*KLHL18*	Homozygous *Klhl18^lowf^* mice	Gene correction by HMEJ	AAV9 and AAV-PHP.eB	Hair bundle morphology rescued and improved auditory function up to 6 months post-treatment	[92]
*DFNA22*	*Myo6^WT/C442Y^* mice, P0–P2	SaCas9-KKH	AAV-PHP.eB	Improved auditory function up to 5 months post-treatment; increased HC survival and improved hair bundle morphology	[110]
*MYO7A*	Patient-derived iPSCs	Gene correction with SpCas9 and HDR	N/A in vitro study	Edited cells displayed recovered stereocilia morphology and electrophysiological response when differentiated into HC-like cells compared to the differentiated unedited cells	[101]
*MYO15A*	Patient-derived iPSCs	Gene correction with Cas9 and HDR	N/A in vitro study	Editing restored hair bundle morphology and electrophysiological responses in the differentiated cells compared to the unedited differentiated cells	[109]
Neomycin-induced HL	Neomycin-treated mice	Disruption of *HTRA2* gene by SaCas9	AAV2/Anc80L65	Inhibition of OHC apoptosis, improvement of ABR thresholds; measured up to 8 weeks after neomycin treatment; 1.73% editing efficiency in vivo	[111]
*TMC1* autosomal recessive	*Baringo* mice, P1	Cytosine base editing of *Tmc1^Y182C/Y182C^*	Dual Anc80L65 vectors	Restoration of HC bundle morphology, sensory transduction current, partial restoration of hearing with improved ABR threshold but no recovery in DPOAE	[99]
*TRMU*	Patient-derived iPSCs	Gene correction of *TRMU* c.28G>T with SpCas9 and HDR	N/A in vitro study	Edited cells differentiated into HC-like cells more efficiently, with improved stereocilia bundle formation, electrophysiological function, mitochondria count and HC marker expression compared to the unedited cells	[107]
Usher syndrome type 1F	*Pcdh15^R245X^* mice	Adenine base editing of c.733C>T mutation	Dual AAV9-PHP.B vectors	Corrected causative c.733C>T substitution mutation in vivo; however, hearing was not restored	[100]
Usher syndrome type 2A	Patient-derived dermal fibroblasts	Gene correction with SpCas9 and HDR	N/A in vitro study	Correction of c.2299delG deletion mutation in patient dermal fibroblasts	[102]
Usher syndrome type 2A	Patient-derived iPSCs	Gene correction with SpCas9 and HDR	N/A in vitro study	Correction of c.2299delG deletion mutation in patient iPSCs	[112]
Usher syndrome type 2A	Patient-derived iPSCs	Gene correction with eSpCas9 and HDR	N/A in vitro study	Correction of c.2299delG and c.2276G>T mutations in patient iPSCs	[105]

ABR, auditory brainstem response; DPOAE, distortion product otoacoustic emission; HC, hair cell; HDR, homology-directed repair; HMEJ, homology-mediated end joining; iPSCs, induced pluripotent stem cells; OHC, outer hair cell.

## 6. Antisense Oligonucleotides

Antisense oligonucleotides (ASOs) are short, single-stranded nucleotides complementary in sequence to a particular region of a gene that can alter the way that RNA is transcribed, spliced or translated upon binding. There are currently two approaches of nucleotide-based treatments for hearing disorders, namely splice-switching ASOs and RNA interference (RNAi).

### 6.1. Splice-Switching ASOs

The correct splicing of pre-mRNA to a mature mRNA strand is vital to produce functional proteins. There are many splicing sites found throughout the genome with highly conserved motifs to ensure the gene expression is regulated, despite the majority not being used [113]. Typically, these well-defined splice sites are recognized by the spliceosome to excise and ligate exons embedded in introns to form mature mRNA. However, mutations that interfere with authentic splicing can cause cryptic splice sites as they create a new splice site that is usually not recognized [114]. To suppress and redirect the aberrant splice site, splice-switching ASOs manipulate the splice site by binding complementarily in an antisense orientation to a specific pre-mRNA sequence and interfering with RNA duplex or normal protein–RNA interactions [115]. Splice-switching ASOs create a steric block on the pre-mRNA strand by inhibiting or altering splicing near the site of the deleterious mutations and without interfering with the endogenous gene expression of other associated genes or regulatory elements.

In early in vivo studies, ASO-based therapeutics were targeted by nucleases and rapidly degraded due to their unmodified phosphate backbone and/or sugar ring of the nucleotide [116,117]. Currently, the chemical structure of ASOs can be modified and specified to the intended target site in the body. Improving the binding affinity, cellular uptake and stability are important considerations for in vivo administration. The most common chemical features include substituting the non-bridging oxygen to a sulfur atom on the phosphorothioate backbone and adding a methyl group on the 2′ sugar position for 2′O-methoxyethyl. These modifications provide stability in vivo and resistance against endogenous Rnase-H nucleases, respectively [115,118].

ASO-based treatments have shown promising therapeutic results in preclinical animal models for inner ear diseases. *Ush1c* mice exhibited disrupted stereocilia bundles, resembling clinical presentation of USH patients [119,120]. The first study to describe ASO-29 as a therapeutic treatment for *Ush1c* c.216G>A mutation was described in 2013 in mice where the mutation created a cryptic 5′ splice site in exon 3 of harmonin, resulting in a frameshift mutation producing a truncated protein [121]. ASO-29 blocked the cryptic splice site and promoted authentic splicing that resulted in the full-length *Ush1c* mRNA being transcribed. *Ush1c* mice treated with ASO-29 at different postnatal days showed significant improvement in their hearing ability. Interestingly, the P3 and P5-treated mutant mice were able to respond to 8–16 kHz sounds but not the P10-treated mutant mice, implying that early treatment may improve the hearing outcome [122].

As the expression of harmonin begins at E15 in mice and is the highest during HC differentiation from P4–P16 [123], multiple strategies to deliver ASO-29 at various stages of development have been reported. ASO-29 was administered via transuterine microinjection into the amniotic cavity and directly into the otic vesicle at E13-E13.5 and E12.5, respectively [124]. While significant hearing preservation at 8, 16 and 24 kHz and partially at 32 kHz was demonstrated with intra-otic delivery, the sub-optimal restoration of hearing to a certain extent was observed in *Ush1c* mice treated with an amniotic delivery method. In a related study, the same group corrected the *Ush1c*.216A mutation in mice with ASO-29 to (i) determine the appropriate ASO dosage and age to rescue auditory HCs and (ii) improve on the Lentz et al. [122] study, in which they observed that treatment must be done at earlier stages to rescue inner ear function [125]. Mutant mice received either one (on P1, P5 or P7), two (at P1 and P3) or four (at P1, P3, P5 and P7) doses of ASO-29 by intraperitoneal injection and all showed the expression of the corrected *Ush1c* mRNA transcript. The ABRs showed variation in the thresholds when simulating for the IHC response regardless of single or multiple dose ASO-29 treatments across post-treatment examination at 1, 3 and 6 months. However, the study did not detect the OHC functional activity by DPOAE when mutant mice received ASO-29 treatment at P7 [125]. It was, therefore, postulated that the therapeutic window for treating defective harmonin is short and must be achieved before HC development or in the early postnatal period in Ush1c mice [43,125]. Indeed, a recent follow-up study of direct RWM injection of ASO-29 to the Ush1c^216AA^ mice inner ear at P1 showed a significant improvement in the hearing function and hair bundle morphology when compared to middle ear administration at later postnatal stages [126].

Disease-causing deep intronic mutations can also be targeted by ASOs. One such mutation in *USH2A* (c.7595-2144A>G) creates a splice donor site in intron 40 that causes the insertion of a 152-base-pair pseudoexon (PE40) [127]. This insertion was predicted to result in a premature stop codon (p.Lys2532Thrfs*56) and a truncated usherin protein [127]. This mutation is the second most common USH2A-causing mutation, accounting for 4% of USH2A patients [128]. Two ASOs designed to target the PE40 region have shown promising results in USH2A patient-derived fibroblasts harboring a single copy of this mutation. Both ASOs resulted in a reduction in the PE40-containing transcript and an increase in the correctly spliced transcript. Additionally, the combination treatment with both ASOs had an additive effect and was highly effective at correcting splicing in vitro [128].

Similarly, the full length of *SLC26A4* exon 8 was rescued by ASO-1029 treatment in a homozygous patient-derived peripheral blood mononuclear cell line harboring the c.919-2A>G mutation. Feng et al. [129] also reported a successful correction of the mis-splicing mutation in *Slc26a4* in a humanized mice model via the exon inclusion method. Recently, a study developed a combination of ASOs and the CRISPR-Cas9 system to target the *CLRN1* mutation at the mRNA and genomic DNA level, respectively. Mutations in *CLRN1* resulted in USH3A, which is characterized by progressive hearing impairments. The splicing correction of the *CLRN1* c.256-649T>G mutation was demonstrated when ASOs or CRISPR-Cas9 were administered alone; however, a cocktail of both treatments did not provide any synergistic effect and rather reduced the efficiency of SpCas9 activity for gene correction [130].

### 6.2. RNA Interference (RNAi)

RNA interference (RNAi) is an endogenous cellular mechanism for targeting foreign double-stranded RNA, such as those introduced by RNA viruses [131]. Recently, the RNAi pathway has been touted as a potential therapeutic approach for targeting genes of interest due to its high potency and specificity. RNAi is stimulated by small double-stranded RNA molecules, including the small interfering RNAs (siRNA) or microRNAs (miRNA), which are recognized by the pathway and digested into shorter fragments by the RNase III-like enzyme Dicer. The fragments are conjugated with an Argonaute protein, forming the RNA-inducing silencing complex (RISC), which guides one strand to form double-stranded RNA that is targeted for cleavage, degradation or repression, while the other strand is removed [132].

siRNAs are derived from double-stranded RNA that has been processed by Dicer into short strands of 21 to 30 nucleotides in length. siRNAs activate the RISC by recruiting the Argonaute protein, which cleaves the sense strand while the antisense strand is guided and remains bound to the RISC. The antisense strand-RISC complex is targeted for complementary base pairing to target mRNA, forming double-stranded RNA, and is processed to be further cleaved. The miRNA-mediated RNAi pathway differs from the siRNA pathway but also employs the Dicer and Argonaute proteins. To effectively target a gene of interest for degradation, there are a number of design considerations. For example, increasing the length of synthetic siRNAs is crucial to target intended mRNA. This increases specificity to the target transcript. Moreover, two overhanging nucleotides must be left on the 3′ end for RNAi machinery recognition [132].

New strategies have recently been developed to rescue deafness through animal models using RNAi [133]. There are many studies experimenting with synthetic siRNA and miRNA to stimulate the RNAi pathway as a therapeutic approach to induce gene silencing for otoprotection. For example, silencing the genes involved in apoptosis, inflammation and oxidative stress, including *TNFα*, *NOX3* and *STAT1*, has showed protective effects on HCs and neurons [134,135,136,137].

Moreover, RNAi therapy for autosomal dominant forms of HL could be targeted through the knockdown of the dominant mutant allele, which we have summarized in Table 4. For example, Maeda et al. [138] designed a synthetic siRNA to form a degradable duplex to suppress the dominant R75W mutation in the mouse *GJB2* gene that causes deafness. In DFNA9, a form of dominantly inherited HL, the c.151C>T founder mutation in *COCH* has been shown to cause progressive hearing impairment in the Belgian–Dutch population. The *COCH* gene encodes for cochlin, a highly expressed protein in fibrocytes of the spiral ligament and spiral limbus. An alternative approach to the RNAi-mediated pathway was investigated in targeting mutant *COCH* transcripts with ASOs for directed degradation by the RNase-H1 mechanism. This was achieved when delivered to *COCH* minigene constructs. Their most potent ASO was able to induce a 60% reduction in mutant *COCH* transcripts without affecting the wild-type *COCH* mRNA levels [139].

Recently, an miRNA-based approach, termed miTmc, was used to suppress the mutant allele in *Bth* mice. *Bth* mice received a single trans-RWM injection of either miTmc or miSafe (control treatment) contained in the AAV vector at P0–P2. The mutant mice showed promising results through the significant preservation of the hearing function and improved HC survival for up to 21 weeks after miTmc treatment. The stereocilia bundle formation in miTmc-treated *Bth* mice was preserved when compared to the untreated and miSafe-treated *Bth* mice where severe degeneration was observed [140]. A study on miRNA treatment in *Bth* mice by Yoshimura et al. [141] administered a single dose of miTmc to knock down the mutant allele at P15–P16 and was sufficient to slow the progression of HL. Moreover, they administered miTmc in *Bth* mice at two additional timepoints, P56–P60 and P84–P90, to test the optimal therapeutic window for rescuing hearing. Interestingly, a mild protective effect on hearing was observed for those treated at P56–P60 and, as expected, treatment at P84–P90 failed to prevent the deterioration of hearing.

An alternative strategy to regenerate HCs from SCs was proposed through gene silencing of the *Hes1*/*Hes5* gene complex using siRNA-based antisense therapy [142,143]. Notch signaling is essential for cell fate and the specification of sensory progenitor cells in the inner ear. The activation of Notch signaling initiated by ligand-receptor binding promotes the expression of *Hes1*/*Hes5*, which induces a SC fate by downregulating *Atoh1* and other prosensory genes [144]. Moreover, *Hes1* is consistently expressed in adulthood to maintain the appropriate HC and SC organization in the organ of Corti. Du et al. [142] observed supernumerary IHCs localized in the middle turn of the cochlea by silencing the *Hes1* and *Hes5* genes. Ultimately, the number of p27^Kip1^-positive SCs was reduced, suggesting that the increase in the HC population was compensated by the number of existing SCs.

## 7. Cell Therapy

Stem cells are precursor cells defined by their ability to self-renew, i.e., give rise to new stem cells and differentiate into many cell types. These properties make them an attractive therapeutic option for a variety of diseases, including SNHL. There are several types of stem cells, but the three types commonly used in regenerative studies are: (i) embryonic stem cells (ESCs), (ii) adult stem cells (ASCs) and (iii) induced pluripotent stem cells (iPSCs).

iPSCs reprogrammed from somatic cells were first reprogrammed from mouse embryonic and adult fibroblast cultures in 2006 through treatment with specific transcription factors [145]. These pluripotency-related transcription factors, also known as ‘Yamanaka Factors’ or ‘OSKMs’ (named OCT4, SOX2, c-MYC and KLF4), are sufficient to induce pluripotency in different adult cell types [145,146]. Although the roles of c-MYC and KLF4 remain unclear, OCT4 and SOX2 are crucial for activating and maintaining the genes associated with pluripotency and renewal, while simultaneously repressing differentiation-associated genes [147,148]. KLF4 is thought to promote cell survival by exerting an anti-apoptotic function, while c-MYC is thought to be involved in the regulation of histone acetylation. c-MYC was initially considered to be non-essential in the reprogramming process, but it is now thought that c-MYC and KLF4 work together to maintain the immortality of iPSCs [149,150].

There are several published protocols to induce HC regeneration from stem cells, each with varied success. Li and colleagues were the first to publish a protocol on generating inner ear progenitor cells derived from mouse ESCs, through the expression of HC markers shown by immunostaining and gene expression analysis [151]. Based on the principles of early otic development, a more sophisticated in vitro feeder layer-based protocol involving several growth factors was shown to successfully promote the differentiation of mouse ESCs and iPSCs into mechanosensitive HC-like cells that were responsive to electrophysiological stimulation [152].

Stem/progenitor cell transplantation therapy uses “corrected” or healthy cells derived from patients to replace older or damaged cells. Stem cell transplantation therapy is a favorable approach for personalized medicine as adult skin cells can be non-invasively harvested from patients and reprogrammed into a pluripotent state. This method of therapeutic approach minimizes incompatibility risks of donor/recipient transplants when administered to patients.

Some studies have reported the use of ASCs/inner ear progenitor cells to form HCs or neural-like cells. Chen et al. [153] transplanted otic-like neural progenitor cells into a mouse model of auditory neuropathy and demonstrated significantly improved ABR thresholds by 46% compared to the untreated mice. In a similar study, Barboza et al. [154] used mouse inner ear progenitor stem cells for HC replacement in adult guinea pigs treated with neomycin. Differentiated cells expressing HC marker MYO7A and SC marker p27^Kip1^ were generated from mature organ of Corti-derived cells and transplanted into the inner ear of the neomycin-treated guinea pigs. The transplanted cells localized in the cochlear basilar membrane and improved the auditory thresholds of the treated guinea pigs.

Similarly, Chen et al. [155] transplanted human iPSC-derived HC-like cells into *Slc26a4*-null mice. Urinary cells from a healthy donor were reprogrammed into iPSCs. These human iPSCs were differentiated into HC-like cells and, separately, SGN-like cells, confirmed by the cell fate markers and electrophysiology studies. When co-cultured, the HC-like cells were able to form synapses with mouse SGNs and differentiated SGN-like cells, as indicated by the expression of CtBP2 and SYP. When HC-like cells were injected into the RWM of *Slc26a4*-null mice, a small population were able to migrate to the organ of Corti, implant into the native cells and maintain the MYO7A expression. The native SGNs were able to form synapses with the implanted cells. The implanted HC-like cells demonstrated no risk of tumorigenesis when injected into NOD/SCID mice. However, the HC-like cells failed to implant in the regular organization of native HCs and hearing was not assessed in this study [155].

In another study, stem cell transplantation was used to treat mice with a connexin 30 (*Cx30*) deletion. Healthy human iPSC-derived outer sulcus-like cells were engrafted into E11.5 mouse cochlea to test their capacity for stem cell regeneration. Interestingly, immunohistochemistry staining of the treated cochleae showed that the engrafted cells expressed *Cx30* one week after transplantation [156]. Although *Cx30*-positive cells were detectable in the non-sensory region in the cochlea, only a limited number of engrafted cells were observed in the inner ear. Similar to the study by Lee et al. [157], only 1–3% of the engrafted cells survived.

A study by Takeda et al. [158] investigated whether the ablation of native HCs would improve the engraftment of transplanted HCs into the organ of Corti. Ablation was achieved by treating *Pou4f3^DTR^*^/+^ mice with diphtheria toxin, which specifically deleted HCs. Human ESCs at day seven of otic differentiation were determined to be at the pre-placodal stage and injected in the scala tympani perilymph via the RWM of neonatal *Pou4f3^DTR^*^/+^ mice treated with diphtheria toxin to specifically ablate IHCs. While the treated *Pou4f3^DTR^*^/+^ mice showed a higher migration and engraftment of HC-like cells in the organ of Corti, the quantity of cells successfully engrafted was still low and the mice did not show improvement in the hearing thresholds.

Adult tissue-specific stem cells reside in somatic tissues and can self-renew and differentiate into specialized cell types. Mesenchymal stem cells (MSCs) are adult stromal cells that display multipotency with limited self-renewal and differentiation capacity. Moreover, studies conducted in animals have shown that MSCs can be differentiated into auditory HCs and neural cells [159,160,161,162]. MSCs release a wide spectrum of proteins, growth factors, cytokines and unique extracellular vesicles (EVs) with immunomodulatory properties. These EVs have been shown to have a potential therapeutic role in rescuing cochlear damage [163]. A recent study demonstrated that these EVs promoted the activation of SOD2, an oxidoreductase, to maintain the levels of mitochondrial reactive oxygen species and increase resistance to oxidative damage in a mouse model of NIHL [164]. In a clinical report, a patient received cochlear implants and the left ear also received allogenic umbilical cord-derived MSC-EVs, the purpose of which was to reduce inflammation and improve cochlear health during the implant procedure [165]. The patient did not show any signs of toxicity or adverse reaction in follow-up appointments at 24 months and demonstrated a significant improvement in speech perception compared to the untreated ear.

Using autologous cells from patients reduces the risk of cell rejection by the immune system and is another promising cell therapy approach for HC regeneration. This form of transplantation has already been FDA-approved and is currently in use for other forms of neurodegenerative diseases like amyotrophic lateral sclerosis [166]. Recently, a Phase I/II clinical trial aimed at treating acquired HL in children with autologous MSCs derived from human umbilical cord blood (hUCB) for HC regeneration has been completed (NCT02038972). A single infusion of autologous hUCB in children aged 5 months to 5 years old all demonstrated an improvement in the ABR thresholds by 5 dB. A follow-up of post-infusion treatment showed that 10 out of 11 subjects showed an improvement in verbal language scoring [167].

Two pilot clinical studies by Lee et al. [168] aimed to transplant autologous MSCs in SNHL patients with auditory neuropathy. Two patients aged 55 and 67 years received an infusion of bone-marrow-derived MSCs for three years. No systematic complication was reported but there was no sign of any hearing improvement in either case [168]. These results obtained from human studies showed that cell-based therapy is possible for treating HL but due to the limited evidence, further study will be needed before transitioning into clinical use.

A major obstacle for cell replacement therapy is evading the immune response when grafting cells. One strategy for overcoming this was presented by Andrade da Silva et al. [169], who used genetic modification in human iPSCs. The human leukocyte antigen genes, which have the most significant effect on graft rejection, were knocked out using CRISPR-Cas9 to reduce immunogenicity when delivering two dimensional-differentiated otic neural progenitor cells into immunocompetent mouse cochlea. This resulted in increased survival of the edited human iPSCs following intracochlear transplantation compared to the unedited cells in a mouse model.

Although stem cell-based therapy is on the cutting edge of modern medicine, there are several challenges and risks to mitigate before clinical application. The major concerns relate to the shared features of stem cells and cancer cells, such as the ability to self-renew, the potency of stem cells (pluripotent or multipotent), indefinite growth and a high proliferation rate. The transplanted cells would need to survive and grow after their insertion into the patient cochlea, without interfering with normal hearing. Additional risks such as the formation of tumors, potential stem cell migration to unwanted sites and immune rejection of transplanted stem cells must also be considered. Furthermore, there are a number of factors that would require highly rigorous control and oversight, including the type of stem cells used, their procurement and culturing history, the level of manipulation and administration site associated with stem cell-based therapy and medicinal products.

## 8. Conclusions

This review has provided an overview of therapeutic strategies for the potential treatment of HL. These included gene-based and cell-based therapies in which, as highlighted, the majority in development are focused on preventing and treating HL. Upon the identification of HL, genetic testing may reveal whether a mutation is present in the known causative genes, which also provides information on the natural history and potential systemic associations. It follows, then, that therapeutics could be delivered to a patient before their hearing deteriorates, and maintaining important protein structures and complexes is significantly easier than restoring them.

Unfortunately, for most patients with inherited HL, severe hearing impairment is evident within the first few years of life. Therefore, an essential consideration is the therapeutic window. To have the most therapeutic benefit, the drug would need to be delivered before the causative mutation has compromised the hearing structures, which is impractical in most cases. However, gene therapy trials in animal models have shown some evidence of functional hearing restoration [73,75]. While this has not yet been assessed in human patients, there is definitely hope that these therapies may have a similar restorative effect. Nevertheless, it will be preferable to deliver biological therapies as early as possible to ensure the best possible outcome for patients.

Conversely for patients who have extensive cell death within their organ of Corti, regenerative and cell therapies would be highly applicable to replace HCs and neurons. Cell therapy is becoming increasingly closer to reality as demonstrated by the advances reported in this review. However, the key question of how implanted cells are seeded into the organ of Corti and how some cells survive while most perish remains unanswered. Understanding this mechanism, perhaps starting with a suitable in vitro scaffold modeling the organ of Corti, could help improve the efficacy and survival of cell supplementation therapies for HL. Nevertheless, these seminal studies into inner ear cell therapies have already illuminated a pathway for the eventual replacement of sensory cells into the human cochlea.

## Figures and Tables

**Figure 1 biomedicines-11-03347-f001:**
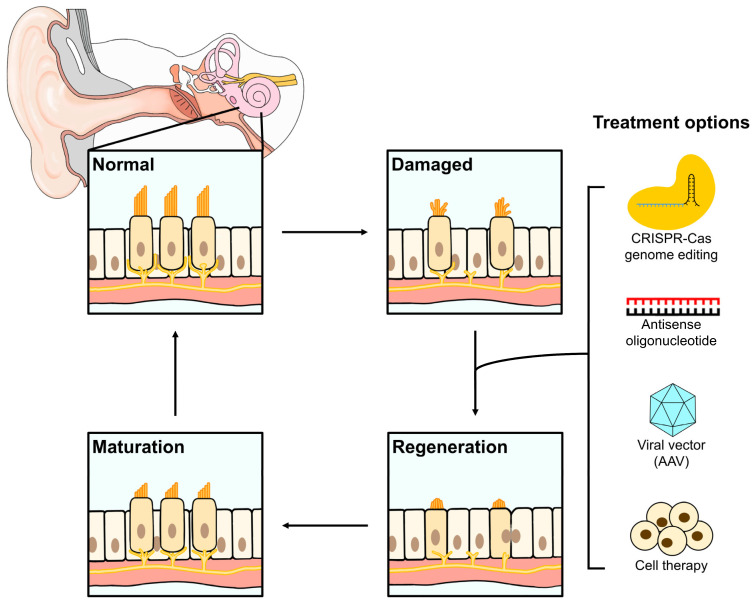
Schematic representation of the sensory HCs and auditory nerves of the cochlea. In the normal organ of Corti, the HCs have upright stereocilia and are innervated with auditory neurons surrounding their base. Various insults (e.g., noise, ototoxic drugs and/or genetic) can damage or cause cell death of the sensory cells resulting in SNHL. Current therapeutic strategies target the signaling pathways and/or alter the gene expression of the affected cells to restore these cells. These approaches can (i) reprogram the cell fate of SCs into cochlear HCs by transdifferentiation, (ii) correct or disrupt the pathogenic gene by genome editing (e.g., CRISPR-Cas systems), modulate gene expression (e.g., ASO), or deliver normal gene transcripts via viral vectors (e.g., AAV), and (iii) use cell-based therapy in combination with gene targeted therapy to replace the damaged HCs and nerves.

**Table 1 biomedicines-11-03347-t001:** Current clinical trials of drug interventions for inner ear hearing disorders.

Cause of Hearing Loss	Intervention (Drug Name)	Route of Administration	Sponsor (Company)	Drug Component (Mechanism of Action)	Study Phase and Status (Clinical Trial ID)
ARHL	Small molecule (PF-04958242)	Oral solution	Biogen (Cambridge, MA, USA)	AMPA receptor potentiator	Phase 1 completed (NCT01518920)
ARHL, tinnitus	Small molecule (AUT00063)	Oral administration	Autifony Therapeutics Limited (Stevenage, UK)	Kv3 potassium channel modulator	Phase 2 completed (NCT02315508; NCT02345031; NCT02832128)
AIED	Biologics (Anakinra (Kineret))	Subcutaneous injection	Northwell Health (New Hyde Park, NY, USA)	Interleukin-1 receptor antagonist	Phase 2 (NCT03587701)
AIED	Monoclonal antibody (Gevokizumab)	Subcutaneous injection	XOMA Corporation (Emeryville, CA, USA)	Proinflammatory interleukin-1 inhibitor	Phase 2 completed (NCT01950312)
AIED	Fusion protein (Rilonacept)	Subcutaneous injection	Stanley Cohen (Regeneron Pharmaceuticals) (Tarrytown, NY, USA)	Interleukin-1 receptor antagonist	Phase 1 (NCT02828033)
DIHL (aminoglycoside)	Small molecule (ORC-13661)	Oral administration	Kevin Winthrop (Oricula Therapeutics) (Seattle, WA, USA)	Mechanotransduction channel blocker	Phase 2 (NCT05730283)
DIHL (cisplatin)	Small molecule (DB-020)	Intratympanic	Decibel Therapeutics (Boston, MA, USA)	Sodium thiosulfate—cation chelator and antioxidant	Phase 1 completed (NCT04262336)
DIHL (cisplatin)	Small molecule	Intravenous	Hyunseok Kang (NRG Oncology) (Philadelphia, PA, USA)	Sodium thiosulfate—cation chelator and antioxidant	Phase 2 completed (NCT04541355)
DIHL (cisplatin)	Small molecule	Intravenous infusion	Sunnybrook Health Sciences Centre (Toronto, ON, Canada)	Sodium thiosulfate—cation chelator and antioxidant; Mannitol—JNK pathway inhibitor	Phase 2 (NCT05129748)
DIHL (cisplatin)	Small molecule	Intratympanic	Sunnybrook Health Sciences Centre (Toronto, ON, Canada)	N-acetylcysteine—antioxidant, reduce inflammation	Phase 1/2 (NCT04291209)
DIHL	Small molecule (SPI-3005)	Oral administration	Sound Pharmaceuticals (Seattle, WA, USA)	Ebselen and allopurinol—mimic antioxidant property of glutathione peroxidase	Phase 2 (NCT01451853; NCT02819856)
Congenital CMV HL	Antiviral (Valganciclovir)	Oral solution	Dr. Ann C.T.M. Vossen (Stichting Nuts Ohra (Amsterdam, The Netherlands) and Leiden University Medical Center (Leiden, The Netherlands))	Viral DNA synthesis inhibitor	Phase 3 terminated (NCT01655212); Phase 3 completed (NCT02005822)
Congenital CMV HL	Antiviral (Valganciclovir)	Oral solution	Albert Park (Genentech) (South San Francisco, CA, USA)	Viral DNA synthesis inhibitor	Phase 2 (NCT03107871)
Congenital CMV HL	mRNA vaccine (mRNA-1647)	Intramuscular injection	ModernaTX, Inc. (Cambridge, MA, USA)	Prophylactic vaccine containing six mRNA sequences for two CMV antigens (glycoprotein B and pentameric glycoprotein complex)	Phase 1 completed (NCT05105048); Phase 2 (NCT0683457); Phase 3 (NCT05085366)
MD, NIHL, tinnitus	Small molecule (SPI-1005)	Oral administration	Sound Pharmaceuticals (Seattle, WA, USA)	Ebselen—mimic antioxidant property of glutathione peroxidase	Phase 1 completed (NCT01452607); Phase 1/2 completed (NCT02603081); Phase 2 (NCT02779192); Phase 2 completed (NCT01444846; NCT03325790); Phase 3 (NCT04677972)
MD	Small molecule (Latanoprost)	Intratympanic	Synphora AB (Uppsala, Sweden)	Prostaglandin F2 α receptor agonist	Phase 2 (NCT01973114)
MD	Small molecule (Montelukast)	Oral administration	House Ear Institute (Merck) (Los Angeles, CA, USA)	Leukotriene receptor antagonists	Phase 4 (NCT04815187)
NIHL	Small molecule (Zonisamide)	Oral administration	Gateway Biotechnology (St. Louis, MO, USA)	Voltage-dependent sodium channel inhibitor and neurotransmitters degrader	Phase 2 (NCT04768569); Phase 2 terminated (NCT04774250)
NIHL, SNHL	Small molecule (NHPN-1010)	Oral administration	Otologic Pharmaceuticals (Oklahoma City, OK, USA)	Disufenton—nitrone-based antioxidant and neuroprotective; N-acetylcysteine—free radical scavenger	Phase 1 completed (NCT02259595)
NIHL (with mitochondrial point mutation)	Small molecule (Vincerinone^TM^ (EPI-743))	Oral administration	Edison Pharmaceuticals (Mountain View, CA, USA)	Vatiquinone—mitochondrial redox modulator	Phase 2 completed (NCT02257983)
SNHL	Small molecule (AM-111)	Intratympanic	Auris Medical (Basel, Switzerland)	Brimapitide—JNK pathway inhibitor	Phase 2 completed (NCT00802425); Phase 3 completed (NCT02561091); Phase 3 terminated (NCT02809118)
SNHL	Small molecule (PIPE-505)	Intratympanic	Pipeline Therapeutics (San Diego, CA, USA)	HC regeneration via Notch signaling pathway and γ-secretase inhibition; synaptic regeneration via netrin/DCC pathway inhibition	Phase 1/2 completed (NCT04462198)
SNHL	Cell-based therapy (Autologous bone marrow infusion)	Intravenous infusion	James Baumgartner, CBR Systems, Inc. (Los Angeles, CA, USA)	Umbilical cord-derived mesenchymal stem cells	Phase 1/2 completed (NCT02038972)
Idiopathic SSNHL	Steroid (AC102)	Intratympanic	AudioCure Pharma GmbH (Berlin, Germany)	Prednisolone	Phase 2 (NCT05776459)
SSNHL	Steroid	Intratympanic	Massachusetts Eye and Ear Infirmary (Boston, MA, USA)	Dexamethasone sodium succinate and prednisone	Phase 3 completed (NCT00097448)
SSNHL	Steroid	Intratympanic	University Hospital Tuebingen (Tuebingen, Germany)	Dexamethasone-dihydrogenphosphate	Phase 3 completed (NCT00335920)
SSNHL	Biologics (Ancrod)	Intravenous	Nordmark Arzneimittel GmbH & Co. KG (Uetersen, Germany)	Anticoagulant—fibrinogen inhibitors and plasminogen activator stimulants	Phase 1/2 completed (NCT01621256)
SSNHL	Steroid	Oral administration	University of Colorado, Denver (Denver, CO, USA)	Dexamethasone and prednisone	Phase 2 completed (NCT03255473)
SSNHL	Steroid	Oral administration	Beijing Tsinghua Chang Gung Hospital (Beijing, USA)	Methylprednisolone hemisuccinate and ginkgo biloba	Phase 4 (NCT04192656)
SSNHL	Steroid	Intratympanic with hydrogel gel	Seoul National University Hospital (Seoul, South Korea)	Dexamethasone and hyaluronic acid	Phase 1/2 (NCT04766853)
SSNHL	Steroid	Oral administration	Northwestern University (Evanston, IL, USA)	Dexamethasone and methylprednisolone	Phase 4 (NCT04826237)
SSNHL	Small molecule (STR001)	Intratympanic versus oral administration	Strekin AG (Basel, Switzerland)	Peroxisome proliferator-activated receptor-γ agonist	Phase 3 completed (NCT03331627)
SSNHL	Small molecule (SENS-401)	Oral administration	Sensorion (Montpellier, France)	R-azasetron besylate—5-HT3 antagonist and calcineurin inhibitor	Phase 2 (NCT05258773; NCT05628233); Phase 2/3 completed (NCT03603314)
SSNHL	Steroid (HY01)	Intratympanic	Heyu (Suzhou, China) Pharmaceutical Technology Co. (Wuxi, China)	Dexamethasone sodium phosphate	Phase 1 (NCT04961099)
Tinnitus	Small molecule (Brexanolone)	Intravenous	Sage Therapeutics (Cambridge, MA, USA)	Gamma-aminobutyric acid A receptor positive modulator	Phase 2 (NCT05645432)
Tinnitus	Small molecule (Gabapentin)	Oral administration	Islamic Azad University of Mashhad (Mashhad, Iran)	Gamma-aminobutyric acid analogue	Phase 2 completed (NCT00555776)
Tinnitus	Small molecule (BGG492A (Selurampanel))	Oral administration	Novartis Pharmaceuticals (Basel, Switzerland)	AMPA-type glutamate receptor antagonist	Phase 2 completed (NCT01302873)
Tinnitus	Small molecule (Etanercept–Enbrel)	Systemic injection	Wayne State University (Detroit, MI, USA)	TNF-α inhibitor	Phase 2 (NCT04066348)
Genetic (otoferlin-mediated HL)	Gene therapy (AK-OTOF)	Intracochlear	Akouos, Inc. (Boston, MA, USA)	Replace defective *OTOF* with healthy *OTOF* cDNA transcript	Phase 1/2 (NCT05821959)
Genetic (otoferlin-mediated HL)	Gene therapy (DB-OTO)	Intracochlear	Decibel Therapeutics (Boston, MA, USA)	Replace defective *OTOF* with healthy *OTOF* cDNA transcript	Phase 1/2 (NCT05788536)
Genetic (otoferlin-mediated HL)	Gene editing (HG205)	Intracochlear	HuidaGene Therapeutics Co., Ltd. (Shangai, China)	CRISPR/Cas13 RNA base editing for p.Q829X mutation in the *OTOF* gene	Phase 1 (NCT06025032)
Genetic (Wolfram syndrome)	Small molecule (Depakine)	Oral administration	Centre d’Etude des Cellules Souches (Corbeil-Essonnes, France)	Sodium valproate—voltage-gated ion channel blocker and histone deacetylase inhibitor	Phase 2 (NCT04940572)
Genetic (mitochondrial DNA tRNA mutation)	Small molecule (KH176)	Oral administration	Khondrion BV (Nijmegen, The Netherlands)	Intracellular redox modulating agent	Phase 2 (NCT04604548)

Note: The database search was conducted as of 18 October 2023, using the key terms (hearing disorder), (sensorineural hearing loss) and (inner ear). Abbreviations: AIED, autoimmune inner ear disease; ARHL, age-related hearing loss; CMV HL, cytomegalovirus-infection-induced hearing loss; DIHL, drug-induced hearing loss; HC, hair cell; MD, Ménière’s disease; NIHL, noise-induced hearing loss; SSNHL, sudden SNHL.

**Table 2 biomedicines-11-03347-t002:** Preclinical therapeutic pipeline for inner ear hearing disorders.

Cause of Hearing Loss	Therapeutic Effect	Intervention (Drug Name)	Route of Administration	Sponsor (Company)	Drug Component (Mechanism of Action)
ARHL, NIHL	Otoprotection	ACOU082	Transdermal patch	Acousia Therapeutics (Tuebingen, Germany)	KCNQ4 receptor agonist
DIHL (cisplatin)	Otoprotection	ACOU085	Intratympanic	Acousia Therapeutics (Tuebingen, Germany)	KCNQ4 receptor agonist
Genetic (*CLRN1*-mediated HL)	Gene therapy	AK-CLRN1	Intratympanic	Akouos, Inc. (Boston, MA, USA)	AAV *CLRN1* gene transfer to cochlear HCs
Genetic (autosomal dominant)	Gene therapy	Undisclosed	N/A	Akouos, Inc. (Boston, MA, USA)	N/A
Genetic (*GJB2*-mediated HL)	Gene therapy	GJB2	N/A	Akouos, Inc. (Boston, MA, USA)	*GJB2* gene correction in SCs
Genetic	HC regeneration	Undisclosed	N/A	Akouos, Inc. (Boston, MA, USA)	N/A
Genetic (vestibular schwannoma)	Gene therapy	AK-antiVEGF	N/A	Akouos, Inc. (Boston, MA, USA)	N/A
AIHL, DIHL, NIHL	Otoprotection	AP-001	Transtympanic	Anida Pharma (Cambridge, MA, USA)	Neuroprotectin (*D1/NPD1*) with anti-inflammatory, cell survival and tissue repair properties
Genetic (*GJB2*-mediated HL)	Gene therapy	AAV.103	N/A	Decibel Therapeutics (collaboration with Regeneron)	AAV *GJB2* gene transfer to cells that would normally express GJB2
Genetic (*STRC*-mediated HL)	Gene therapy	AAV.104	Undisclosed	Decibel Therapeutics (collaboration with Regeneron)	AAV *STRC* gene transfer to selectively target outer HCs
Genetic	Gene therapy	AAV.105	Undisclosed	Decibel Therapeutics (Boston, MA, USA)	Undisclosed
Genetic	Regeneration	AAV.201	Undisclosed	Decibel Therapeutics (Boston, MA, USA)	ATOH1 and SOX2 inhibitor
Genetic	HC regeneration	Undisclosed	Undisclosed	Decibel Therapeutics (Boston, MA, USA)	Undisclosed
Tinnitus	Undisclosed	GW-TT2	Nasal spray	Gateway Biotechnology (St. Louis, MO, USA)	FDA-approved drug
Tinnitus	Gene therapy	GW-TT5	N/A	Gateway Biotechnology (St Louis, MO, USA)	N/A
Tinnitus	Undisclosed	GW-TT23	Nasal spray	Gateway Biotechnology (St. Louis, MO, USA)	N/A
DIHL, NIHL	Neural protection and regeneration	HB-097 (Cometin)	N/A	Hoba Therapeutics (Copenhagen, Denmark)	Neurotrophic factor targeting the JAK-STAT3 and MEK-MERK pathways promote the survival of neuronal cells
MD	Otoprotection	IB2000	N/A	IntraBio (London, UK)	Diferuloylmethane analogue with anti-inflammatory property
MD	Otoprotection	IB5000 (Betahistidine)	N/A	IntraBio (London, UK)	Monoamine oxidase inhibitor
Auditory neuropathy	Neural cell replacement therapy	ANP1	N/A	Lineage Cell Therapeutics (Carlsbad, CA, USA)	Replace damaged cells with auditory neuronal progenitors
Genetic (*TMPRSS3*-mediated HL)	Gene therapy	Myr-201	N/A	Myrtelle, Inc. (Wakefield, MA, USA)	AAV *TMPRSS3* gene transfer
AIED	Otoprotection	OR-102A	N/A	O-Ray Pharma (Pasadena, CA, USA)	TNFα inhibitor
SNHL	Otoprotection	OR-102C	N/A	O-Ray Pharma (Pasadena, CA, USA)	Protective effect during cochlear implant surgery
ARHL	Otoprotection	OR-112	N/A	O-Ray Pharma (Pasadena, CA, UAS)	N/A
MD	Otoprotection	ORB-202 (Betamethasone)	Intratympanic	Orbis Biosciences (Lenexa, KS, USA)	Fast film-forming agent (FFA) encapsulating betamethasone
SNHL	HC regeneration	Small molecule (OPI-001)	N/A	Otologic Pharmaceuticals (Okhaloma City, OK, USA)	Small molecule drugs and siRNA to promote HC regeneration
Genetic (*TMPRSS3*-mediated HL)	Gene therapy	RHI100	N/A	Rescue Hearing Inc. (Gainesville, FL, USA)	AAV *TMPRSS3* gene transfer
Genetic (non-syndromic HL)	Gene therapy	RHI400	N/A	Rescue Hearing Inc. (Gainesville, FL, USA)	Target major cause of non-syndromic HL
SNHL	Gene therapy	RHI500	N/A	Rescue Hearing Inc. (Gainesville, FL, USA)	Target HL in cochlear implant population
Neurodegenerative disease	Gene therapy	RHI600	N/A	Rescue Hearing Inc (Gainesville, FL, USA)	N/A
Auditory neuropathy	Neural cell replacement therapy	Rincell-1	N/A	Rinri Therapeutics (Sheffield, UK)	Replace damaged cells with embryonic stem cells
Genetic (*OTOF*-mediated HL)	Gene therapy	OTOF-GT (SENS-501)	N/A	Sensorion (Montpellier, France)	Dual AAV *OTOF* gene transfer to cochlear HCs
Genetic (*GJB2*-mediated HL)	Gene therapy	GJB2-GT	N/A	Sensorion (Montpellier, France)	Dual AAV *GJB2* gene transfer to cochlear HCs
SNHL	Regeneration	SPI-5557	N/A	Sound Pharmaceuticals (Seattle, WA, USA)	Cyclin-dependent kinase (p27^Kip1^) inhibitor
DIHL (cisplatin)	Otoprotection	LPT99	Transtympanic (hydrogel formulation)	Spiral Therapeutics (South San Francisco, CA, USA)	APAF-1 inhibitor, anti-apoptotic
DIHL (cisplatin)	Otoprotection	TT001 (AZD5438)	N/A	Ting Therapeutics (Ohama, NE, USA)	Cyclin-dependent kinase (p27^Kip1^) inhibitor
DIHL (cisplatin)	Otoprotection	TT002 (Niclosamide)	N/A	Ting Therapeutics (Ohama, NE, USA)	EGFR-ERK pathway
DIHL	Otoprotection	TT003 (Piperlongumine)	N/A	Ting Therapeutics (Ohama, NE, USA)	Akt phophorylation inhibition via the accumulation of reactive oxygen species

Abbreviations: AIED, autoimmune inner ear disease; ARHL, age-related hearing loss; DIHL, drug-induced hearing loss; HC; hair cell; HL, hearing loss; MD, Ménière’s disease; NIHL, noise-induced hearing loss; SC, supporting cell; SSNHL, sudden SNHL.

**Table 4 biomedicines-11-03347-t004:** Antisense oligonucleotide-based therapy for SNHL in preclinical models.

Gene/Disease Model	Model	Mechanism of Action	Chemical Modification	Effect	Reference
DFNA9	Minigene splicing assay	RNAse H1-mediated degradation with siRNA	2′-deoxyribose flanked with 2′-O-methyl-RNA with PS backbone	Selective suppression of the mutant *COCH* c.151C>T pre-mRNA transcript	[139]
DFNA3	*Gjb2^R75W^* mice, P19	RNAi-mediated degradation with siRNA	2′-deoxythymidine residues	Selective suppression of the dominant negative *Gjb2^R75W^* expression in the organ of Corti and prevention of HL in mice	[140]
Pendred syndrome	Patient-derived PBM cells; chimeric mice, P3–P4	Splice modification	2′-MOE with PS backbone	Correction of splicing caused by *SLC26A4* c.919-2A>G mutation in patient-derived blood cells and in the chimeric mouse inner ear	[129]
DFNA36	*Beethoven* mice, P0–P2	RNAi-mediated degradation with miRNA	miRNA conjugated to AAV2/9 vector	Improved ABR thresholds; expression of *Tmc1^Bth/+^* mutant allele significantly reduced compared to the untreated mutant Bth mice	[140]
DFNA36	*Beethoven* mice, P15–P16, P56–P60 and P84–P90	RNAi-mediated degradation with miRNA	miRNA conjugated to AAV2/9 vector	Single dose of miTmc at P15–P16 showed significant preservation of hearing and HC morphology at 8–12 weeks of age	[141]
Usher syndrome type 1C	*Ush1c*^216AA^ mice, P3–P18	Splice modification	2′-MOE with PS backbone	Improved hearing function and correction of harmonin expression in HCs was maintained up to P180 in *Ush1c* mutant mice injected at P3 and P5	[122]
Usher syndrome type 1C	*Ush1c*^216AA^ mice, P1–P7	Splice modification	2′-MOE with PS backbone	ASO-29 treatment at P1–P5 rescued both IHCs and OHCs but treatment at P7 only partially rescued IHC function	[125]
Usher syndrome type 1C	*Ush1c*^216AA^ mice, P1–P20	Splice modification	2′-MOE with PS backbone	Detectable harmonin protein expression and correction of full-length harmonin transcript; post-treatment of ASO-29 at 1 month showed significant improvement in hearing	[126]
Usher syndrome type 1C	*Ush1c*^216AA^ mice, E12.5	Splice modification	2′-MOE with PS backbone	Partial restoration of hearing and vestibular function was observed 1 month after injecting in utero in E12.5 *Ush1c* mice	[124]
Usher syndrome type 2A	Patient-derived fibroblasts (*USH2A* c.7595-2144A>G and c.12575G>A)	Splice modification	2′-MOE with PS backbone	ASO treatment excluded intron 40 caused by the *USH2A* c.7595-2144A>G mutation into the mature *USH2A* mRNA transcript in patient-derived fibroblasts	[128]
Usher syndrome type 3A	Minigene splicing assay	Splice modification	2′-MOE with PS backbone	ASO treatment showed robust splicing correction of the *CLRN1* c.254-649T>G splicing mutation in the cell lines carrying patient mutation	[130]
Cisplatin-induced HL	Cisplatin-treated rats	RNA-mediated degradation with siRNA	N/A	NOX3 siRNA treatment showed otoprotective effects by suppressing inflammation and apoptotic activity in the cochlea	[135]
Cisplatin-induced HL	Cisplatin-treated rats	RNAi-mediated degradation with siRNA	N/A	Knockdown of *STAT1* inhibited cisplatin-induced inflammation and protected HCs in rat cochleae	[134]
NIHL	Noise-exposed mice	RNAi-mediated degradation with siRNA	N/A	NOX3 expression was reduced in OHCs and hearing threshold shifts were attenuated in noise-exposed mice	[136]
NIHL	Noise-exposed rats	RNAi-mediated degradation with siRNA	N/A	Knockdown of *TNFα* downregulated the genes involved in ROS generation, and therefore improved the hearing outcome after noise exposure	[137]
HC regeneration	Neomycin-treated mouse cochlea explants	RNAi-mediated degradation with siRNA	N/A	*Hes1* and *Hes5* mRNA expression levels were reduced where an increase in *Atoh1* expression level led to HC regeneration	[142]
HC regeneration	Noise-exposed guinea pigs	RNAi-mediated degradation with siRNA	N/A	Supernumerary IHCs were observed in 50% of the cochleae treated with *Hes1* siRNA; ABR thresholds were well preserved	[143]

2-MOE, 2′-O-methoxyethyl; HC, hair cell; HL, hearing loss; IHC, inner hair cell; NIHL, noise-induced hearing loss; OHC, outer hair cell; PBM, peripheral blood mononuclear, PS, phosphorothioate; ROS, reactive oxygen species.

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
