# Peer review of "Recent Therapeutic Progress and Future Perspectives for the Treatment of Hearing Loss"

_biomedicines, 2023, doi:10.3390/biomedicines11123347_

Round 1

Reviewer 1 Report

Comments and Suggestions for Authors

The authors have provided a very balanced, comprehensive, critical and timely review of the wide variety of treatments for sensorineural hearing loss.  The manuscript is well written and summarizes up to date and intriguing sources.  The tables are quite informative and provide lists of recent clinical trials for hearing loss and tinnitus.

Author Response

Comments 1: The authors have provided a very balanced, comprehensive, critical and timely review of the wide variety of treatments for sensorineural hearing loss. The manuscript is well written and summarizes up to date and intriguing sources. The tables are quite informative and provide lists of recent clinical trials for hearing loss and tinnitus.

Response 1: We would like to thank reviewer for the comprehensive review of our manuscript.

Reviewer 2 Report

Comments and Suggestions for Authors

As a general otolaryngologist with an interest in otology and clinical research, I found this manuscript interesting and informative. I cannot say whether the lists of therapies under investigation for various forms of SNHLs the authors provide are exhaustive, but they certainly give a good idea of current research and therapeutic perspectives in the field. In addition, I am not sure if there is space for a more simplified presentation of the crisp method, but I have the feeling that the introduction of this method is not useful in the way it is described in the manuscript.

Author Response

Comments 1:

I am not sure if there is space for a more simplified presentation of the crisp method, but I have the feeling that the introduction of this method is not useful in the way it is described in the manuscript.

Response 1: Minor revisions.

We would like to thank reviewer for the comprehensive review of our manuscript.

We have modified the paragraph of the CRISPR method (Section 5.1, Paragraph 1 and 4, Line 335-413; Section 5.2, Paragraph 1 and 4, Line 414-459).

5. Gene Editing Therapies

5.1. CRISPR-Cas Therapeutics

The CRISPR-Cas9 method of genome editing has great potential to be used for treating certain forms of SNHL. CRISPR, or Clustered Regularly Interspaced Palindromic Repeats, is currently the most prominent method for gene editing. The CRISPR system has two components: a Cas nuclease and guide RNA (gRNA) which guides the nuclease to the target area in the genome [82]. The target area is required to be adjacent to a protospacer-adjacent motif (PAM), a short 3-6 bp sequence found throughout the genome, however recent variants have been identified that recognize much wider PAM sequence varieties and have been dubbed “near PAM-less” [83,84]. Once guided to the target area, the Cas nuclease causes a double-stranded break in the DNA, where modifications can be made alongside repair of the break [85,86].

CRISPR-mediated genome editing is especially applicable for certain genetic hearing disorders. Many genetic hearing disorders arise from single mutations in the DNA sequence of a gene, which can theoretically be rectified using CRISPR. As the target cell types in the cochlea are terminally differentiated, many of the strategies described later involve disrupting the target allele of an autosomal dominant disorder or using base editing to modify and repair single nucleotides. This is as other methods of repair, notably homology-directed repair, require cell division to occur to introduce the modification [87]. At this stage, CRISPR-mediated therapy for some forms of genetic HL are early in development, however more work has recently been produced in this area, which we have summarized in Table 3.

Gao et al. [88] used a CRISPR approach to disrupt the Tmc1 mutation in the Bth mouse model of autosomal dominant HL. The study employed Streptococcus pyogenes Cas9 (SpCas9, one of the most widely used experimentally) to target and disrupt the mutant Tmc1Bth allele on neonatal mice, using liposomes to deliver the gRNA-CRISPR-Cas9 system into the mouse scala media. The CRISPR-mediated therapy mitigated the progressive cochlear HC death exhibited by the Bth model compared to un-injected mice. Treated Bth mice also exhibited improved low-middle frequency ABR thresholds and increased startle response compared to un-injected mice [88]. A more recent study by György et al. [89] used a PAM variant of the Staphylococcus aureus Cas9 (SaCas9-KKH) which was able to target the Bth mutation in the mice and the human p.418K mutation in human cell lines. The effects of this treatment were sustained for longer than the Gao et al. [88] study. Finally, Zheng et al. [90] used an RNAi approach with CRISPR-CasRx, which is an attractive approach as it has decreased risk of off-target effects, with transcripts being targeted rather than the genome, resulting in improved hearing up to 15 weeks post-treatment. Similar studies have been conducted using SpCas9 and SaCas9-KKH on mouse models of KCNQ4-associated autosomal dominant hearing loss [91,92].

A strategy for gene editing by homology-mediated end joining (HMEJ) was recently presented by Gu et al. [93]. A mutation in the Klhl18 (Kelch-like family member 18) gene leads to the dysfunction of IHCs, which results in abnormal ABR but normal DPOAE thresholds [94]. AAV9 and AAV-PHP.eB were used to package the CRISPR-Cas9 knock-in system for delivery into Klhl18lowf mice, which modelled the autosomal recessive condition in humans. Using the HMEJ strategy, they were able to rescue IHC stereocilia morphology eight weeks after injection of the dual AAV system and demonstrated improved auditory function up to six months post-treatment [93].

5.2. Base editing therapeutics

While the sequence-targeting ability of the Cas nuclease is highly useful, double-stranded breaks are not usually practical for correction of mutations, especially for single base point mutations where swapping of one nucleotide pair is sufficient. Double-stranded breaks induced by Cas additionally induce DNA repair mechanisms that may either create insertion/deletion polymorphisms (indels), leading to off-target effects, or reintroduction of the original pathologic variant [95,96]. The most notable innovation in CRISPR technology to overcome this is the introduction of base editors, which have recently been tested in preclinical models of genetic SNHL.

Base editors are able to modify individual base pairs in double-stranded DNA without causing a double-stranded break. They are chimeric proteins consisting of a modified Cas nuclease, where one of the catalytic domains is inactivated, and a single-stranded DNA deaminase enzyme [85,97]. Base editors are advantageous for gene editing HCs and SCs as mitosis is not required to complete changes to the genome [98]. There are two main classes of base editors, cytosine base editors which convert C•G base-pairs to A•T, and adenosine base editors which catalyze the inverse A•T to C•G conversion.

Two papers by Yeh and colleagues were the first to demonstrate the potential of base editing in inner ear therapy. The first sought to modify the β-catenin protein to prevent its degradation by the proteasome. β-catenin is a key effector protein in the Wnt signaling pathway, a crucial pathway regulating HC development, and is targeted for degradation upon phosphorylation by GSK3β. The study by Yeh et al. [99] showed modification of β-catenin prevented phosphorylation and degradation and prolonging Wnt signaling in vitro. Moreover, the authors used a lipid-mediated approach for delivery, as in Gao et al. [88], and showed specific gene editing in SCs when the base editor was applied in vivo. The second paper showed some improvement in hearing when a base editing approach was applied to treat Baringo mice, which have an autosomal recessive form of HL caused by a T•A-to-C•G mutation leading to a non-functional TMC1 protein [100]. This particular approach additionally highlights the advantage of base editing in treating autosomal recessive disorders by being able to specifically correct the mutation at both alleles in terminally differentiated cells.

Very recently, base editing was applied to an animal model of USH type 1 [101]. This study marked the first in vivo application of CRISPR for USH. The authors generated a mouse model of USH1F, first using CRISPR to introduce the R245X (c.733C>T) mutation which produces a stop codon in place of an arginine, resulting in a truncated Pcdh15 protein and disrupted hair bundles. The authors used a dual AAV vector with a split-intein system to deliver adenine base editors to the cochlea of Pcdh15R245X/R245X mice at P0-P1 and a late-stage conditional knockout model where Pcdh15 is deleted at P15.5 under the control of the Myo15-Cre promotor. The adenine base editors were successful in correcting the C>T mutation for A>G in vivo, however were unable to restore hearing in the Pcdh15R245X/R245X mice [101]. The base editor was able to restore hearing in the late-stage conditional knockout model however, indicating the therapy may potentially be more successful if administered in the prenatal period.